# OpenReview forum: "OrthAlign: Orthogonal Subspace Decomposition for Non-Interfering Multi-Objective Alignment"
_ICLR.cc/2026/Conference — ICLR 2026 Poster_

### Official Review · Reviewer_q8r6 · 2025-10-29

**Soundness:** 2
**Presentation:** 2
**Contribution:** 2
**Rating:** 4
**Confidence:** 3

**Summary:**

This paper tackles a specific problem in LLM alignment: resolving conflicts between multiple competing preferences. The core idea of using orthogonal subspace decomposition to create non-interfering parameter updates is novel and intuitive. The empirical results presented are strong, showing consistent advantages over a variety of baselines in balancing objectives and preserving the distributions of prior preferences.

However, these strengths are severely undermined by significant, disqualifying weaknesses in the paper's current form. The paper is exceptionally difficult to read and understand. The paper's core theoretical guarantees (Theorems 1 & 2)  are flawed.

**Strengths:**

1. The paper's key insight is to reframe the MPA conflict from a data or reward-level problem to a parameter-level problem of gradient interference. This is an elegant and promising direction for alignment research.

2. OrthAlign demonstrates SOTA performance across all reported experiments, outperforming 7+ baselines.

**Weaknesses:**

1. The final, practical algorithm is not presented in one clear, understandable place. The reader is forced to piece it together from a high-level description in Sec 3.1, a projection formula in Sec 3.3, and a critical but separate "Adaptive Subspace-Rank Selection" algorithm in Appendix B.1. This makes the method extremely difficult to follow and reproduce.

2. The paper is critically ambiguous about what matrix the SVD is performed on. Algorithm 1 (App B.1) states it is performed on the full parameter matrix $W$. However, the theory in Sec 3.1 is built around the LoRA adaptation matrix $\Delta W$. These are completely different matrices with vastly different computational and theoretical implications, and this ambiguity is unacceptable. This paper involves very complicated linear algebra-related notations, but the authors didn't pay much attention to making all the nottations clearly defined and consistent throughout the paper.

3. The paper reuses the symbol $\tau$  for two completely different concepts, which is a major source of confusion. It is used to mean (1) the spectral norm bound in Theorem 2, and (2) the reward tolerance threshold in Algorithm 1 and Eq. 8.

4. My major concern is for the significance of the theorems. From my perspective, the inequalities in Theorem 1 and Theorem 2 hold true in general when Assumption 3.1 holds. They are not new results for the particular setting considered in this paper.  This is also verified by the the proofs provided in Appendix F.  They are direct, first-principles applications of standard linear algebra tools: the Rayleigh quotient for Theorem 1, and the triangle inequality and orthogonality for Theorem 2. While correct (assuming the premises), the proofs themselves contain no technical novelty. The contribution is in the setup, not the derivation.

5. The provided theorems, even if they were verifiable, only offer guarantees at the parameter-weight level. Theorem 1 bounds the second-order parameter change, and Theorem 2 bounds the parameter matrix norm. Neither theorem provides a direct guarantee on the actual, high-level performance metric (i.e., the reward $\mathcal{R}(W; X_{safe})$). The link between "parameter norm stability" and "reward score stability" is non-trivial for a deep, non-linear model and is never formally established.

6. We have no idea of how large $\tau$ is in Theorem 2. Thus, the practical algorithm (Algorithm 1) must fall back on an empirical reward-checking loop to find a safe rank $k^*$. This implicitly admits that the theory (Theorems 1 & 2) is insufficient on its own to guarantee performance, and a heuristic search is still required.

7. Lemma 1 relies on $\nabla g(\theta) \in \mathcal{S}_k$, but the paper fails to provide a justification for why the safety gradient ($\nabla g(\theta)$) would lie within the principal subspace ($\mathcal{S}_k$) of the safety curvature Hessian ($H_s$).

**Questions:**

Please see the weakness above.

---

> ### Author Response · Authors · 2025-11-21
> **Author Response to Reviewer q8r6 [1/3]**
>
> We would like to express our deepest respect for your review! In response to your comments, we have carefully prepared a point-by-point reply:
>
> ---
>
> > `Weakness1` :  Fragmentation of the practical algorithm presentation
>
> **A1.** We sincerely thank  you for this insightful and constructive feedback. We agree that presenting the practical algorithm in one more clear, understandable place would strengthen our paper's overall quality and presentation.  We have completely restructured **Section 3**
>
> 1. **Unified Practical Framework (Section 3.1):** We have consolidated the entire execution workflow into **Section 3.1 (The OrthAlign Framework)**. This section now logically guides the reader through the three core steps of our method:
>    - **Step 1:** Orthogonalized Preference Decomposition (introducing the decomposition formula).
>    - **Step 2:** Adaptive Subspace-Rank Selection (explaining the insight of minimizing impact on prior preferences and integrating the core pseudocode directly here).
>    - **Step 3:** Subspace-Constrained Optimization (presenting the final projection and update rule).
> 2. **Theoretical Analysis (Section 3.2):** We have moved the theoretical proofs and stability bounds to **Section 3.2**. This "Methodology-First, Theory-Second" structure ensures that readers can first grasp *how* to implement OrthAlign before diving into the mathematical guarantees of *why* it works.
>
> We believe this reorganization directly addresses  your feedback and significantly enhances the readability and reproducibility of our work.
>
> ----
>
> > `Weakness2 & Weakness3` : Ambiguity in SVD matrix definition and inconsistent symbol usage
>
> We sincerely thank you for your  examination of our mathematical notations. To ensure mathematical precision and clarity, we have enforced a stricter distinction between the full parameter matrix and the LoRA adaptation matrix throughout the manuscript. Furthermore, we have refined our notation to systematically distinguish the symbols for spectral norm bounds and reward tolerance thresholds.
>
> In response to your valuable feedback, we have carefully revised our manuscript to ensure absolute clarity and consistency:
>
> 1. **Standardized SVD Target:** We have explicitly standardized the notation throughout the paper (including Algorithm 1 and Section 3.1) to state that the SVD is performed strictly on the **LoRA adaptation matrix ($\Delta W$)**, keeping it distinct from the full weights $W$.
> 2. **Resolved Symbol Conflict:** We have introduced distinct symbols to eliminate the overlap you identified. We now exclusively use **$\gamma$** to denote the **reward tolerance threshold**, ensuring it is clearly distinguished from the spectral norm bound ($\tau$) used in Theorem 2.
>
> These definitions have been updated and highlighted in the **revised PDF** to reflect this consistency.
>
> **Updated Algorithm Schematic:**
>
> **Input:**
>
> - Current LoRA adaptation matrix **$\Delta W$**
> - Safe input set $X_{\text{safe}}$
> - Reward Tolerance **$\gamma$** *(Distinct Symbol)*
> - Maximum rank $r_{\max}$
>
> **Output:**
>
> - Optimal Rank $r^\star$
>
> ------
>
> Step 1: Compute Singular Value Decomposition
>
> Perform SVD strictly on the LoRA matrix:
>
> $$\Delta W = U \Sigma V^\top$$
>
> Step 2: Initialize
>
> Set $l \leftarrow 8, \quad h \leftarrow r_{\max}, \quad r^\star \leftarrow 0$
>
> Calculate baseline reward: $R_{\text{base}} \leftarrow \mathcal{R}(\Delta W; X_{\text{safe}})$
>
> Step 3: Binary Search for Optimal Rank
>
> While $l \le h$ do:
>
> 1. Set candidate rank: $k \leftarrow \lfloor (l + h)/2 \rfloor$
>
> 2. Calculate noise magnitude: 2. Calculate noise magnitude: 2. Calculate noise magnitude: $\hat{\sigma}$= Average(σ_1, ..., σ_r)
>
> 3. Reconstruct temporary matrix: $\Delta W_{\text{new}} = U \hat{\Sigma}^{(k)} V^\top$
>
>    (Inject $\hat{\sigma}_{\text{noise}}$ into the trailing (last) $k$ singular values)
>
> 4. Check Tolerance:
>
>    If the reward degradation is within $\gamma$:
>
>    $$\big| \mathcal{R}(\Delta W_{\text{new}}; X_{\text{safe}}) - R_{\text{base}} \big| \le \gamma$$
>
>    Then:
>
>    - $r^\star \leftarrow k$  *(Store valid rank)*
>
>    - $l \leftarrow k + 1$ (Try larger subspace / more compression)
>
>      Else:
>
>    - $h \leftarrow k - 1$ *(Shrink subspace)*
>
> Step 4: Return
>
> Return $r^\star$

---

> ### Author Response · Authors · 2025-11-21
> **Author Response to Reviewer q8r6 [2/3]**
>
> > `Weakness4` :  Concern is for the significance of the theorem 1 and 2
>
> **A4:**  Thank you for your comment! We would like to clarify that our primary contribution lies in proposing a new paradigm: mitigating preference alignment conflicts via constrained gradient updates at the parameter level. This is exactly what we emphasized in our **Introduction** rather than claiming theoretical novelty in the mathematical derivations themselves. However, we believe these theorems provide essential theoretical guidance to ensure that our parameter updates remain efficient and stable.
>
>
>
> ------------------------------------
>
>
>
> > `Weakness5.` The theorems verify parameter-level stability but fail to formally establish the non-trivial link to actual reward score stability in deep, non-linear models. && `Weakness6.` Reliance on heuristic search.
>
> **A5&A6:** Thank you for your comment. We appreciate the opportunity to clarify how our theoretical framework directly informs our practical algorithm to ensure both high performance and stability. Rather than being a heuristic fallback, our **Adaptive Subspace-Rank Selection** functions as a rigorous simulation grounded in the spectral properties of the model.
>
> 1. **Theoretical Foundation and "Increment" Interpretation**
>
> Fundamentally, we posit that the capability increment for specific tasks is constituted by singular vectors and their corresponding singular values. In the context of Parameter-Efficient Fine-Tuning (LoRA) preference alignment, learned behaviors are primarily encoded by a few dominant directions and their associated singular values. Consequently, Theorems 1 and 2 provide the necessary condition: preventing parameter updates within these specific vector spaces significantly mitigates interference with previously acquired capabilities.
>
> 2. **Fine-Grained Exploration of Theorems 1 & 2**
>
> Based on this intuition, our Adaptive Rank Selection serves as a fine-grained practical extension of Theorems 1 and 2. Taking a 7B model with a hidden dimension of 4,096 as an example, masking the top-16 singular components (critical for safety) leaves a vast update space of 4,080 dimensions. In this orthogonal subspace, singular values approach zero (typically around $10^{-8}$ due to machine precision), whereas the top-16 eigenvalues are orders of magnitude larger.
>
> 3. **Simulating Impact via Noise Injection (Addressing Eq. 7)**
>
> We aim to rigorously constrain the update rank because, as analyzed in Eq. 7, parameter updates within the remaining 4,080 dimensions can still introduce spectral increments that may interfere with the model's behavior:
>
> $$\sum_{j=r+1}^{\max(m,n)} \sigma_j v_j^T X \approx 0 \xrightarrow{\text{update}} \sum_{j=r+1}^{\max(m,n)} \tilde{\sigma}_j \hat{v}_j^T X \neq 0$$
>
> Therefore, our **low-rank selection** mechanism essentially regulates the permissible range of these increments. We achieve this through a **simulation process akin to noise injection** in traditional machine learning: we proactively amplify the near-zero singular values to match the magnitude of the top-16 components. By constructing this perturbed matrix, we simulate the maximum potential impact of future updates. This allows us to rigorously determine a safe subspace rank that guarantees stability without the need for expensive full re-training.
>
> **Time Analysis**. Assuming a practical lower bound for the rank is $r=8$ (which is already relatively low in practice), the binary search converges very quickly.   To demonstrate this efficiency, we conducted a test on 1 GPU with right boundary ranks set to 16, 24, and 32, respectively. We repeated the search process  5 times  for each boundary setting. As shown in **Table B** below, the average runtime for the entire adaptive selection process is approximately  2 to 3 minutes. This indicates that our method is computationally efficient and does not impose a significant overhead for practical use. We have included the corresponding experimental results in Appendix B.3.
>
> Table B.  Average time consumption
>
> | **Boundary Index (Right Limit)** | **Average Time (second)** | **Variance (second)** |
> | -------------------------------- | ------------------------- | --------------------- |
> | **16**                           | 126.13                    | 0.67                  |
> | **24**                           | 190.77                    | 1.72                  |
> | **32**                           | 189.84                    | 6.28                  |
>
> **In summary, our reward selection strategy enables the rapid identification of synthetic rank updates in a low-rank environment , efficiently exploiting the monotonic performance impact induced by noise interference to quickly locate the optimal rank.**

---

> ### Author Response · Authors · 2025-11-21
> **Author Response to Reviewer q8r6 [3/3]**
>
> > `Weakness7`  Why the safety gradient would lie within the principal subspace  of the safety curvature Hessian .
>
> **A7.** Thank you for your question. To address your concern, we would like to clarify that this relationship is structurally enforced by our method's design.
>
> We first compute the principal subspace (approximating the safety curvature Hessian) specifically to capture the dominant gradient directions of the safety objective. Subsequently, as rigorously defined in **Equation 10**, our algorithm **defaults to projecting** any parameter updates for the next preference onto the orthogonal complement of this calculated subspace:
>
> $$\Delta W_{\text{new}} = P \cdot \nabla_W \mathcal{L}_{\text{new}}(W)$$
>
> **This projection mechanism explicitly ensures that the new updates would lie within the principal subspace  of the safety curvature Hessian .**
>
> ---
> We are very grateful for this detailed list of suggestions, which will significantly improve the paper's quality and readability. If any concerns remain, or if new questions arise, we would be grateful to hear from you.

---

> > ### Author Response · Authors · 2025-11-25
> > **Thank you & Looking forward to further discussion！**
> >
> > **Dear Reviewer q8r6,**
> >
> > We sincerely thank you for your detailed and critical review. We appreciate that you recognized the **"elegance"** of our core insight and acknowledged our **SOTA performance** against 7+ baselines. We have taken your concerns regarding presentation and notation very seriously and have extensively revised the manuscript to address them. **Below is a concise point-by-point summary of our rebuttal:**
> >
> > > `Weakness 1:` Fragmentation of the practical algorithm presentation.
> >
> > **We have completely restructured Section 3 to follow a "Methodology-First" approach:**
> > * **Unified Framework (Sec 3.1):** We consolidated the workflow into one coherent section: (1) Decomposition, (2) Adaptive Rank Selection, and (3) Subspace-Constrained Optimization.
> > * **Theory (Sec 3.2):** We moved proofs and bounds to a dedicated section *after* the method, ensuring readers understand *how* OrthAlign works before diving into *why* it is stable.
> >
> > > `Weakness 2 & 3:` Ambiguity in SVD matrix definition ($W$ vs $\Delta W$) and inconsistent symbols ($\gamma$ vs $\tau$).
> >
> > We have rigorously standardized all notation in the revised paper and Algorithm 1:
> > * **Target Matrix:** We explicitly state that SVD is performed strictly on the **LoRA adaptation matrix ($\Delta W$)**, distinct from the full weights ($W$).
> > * **Symbol Resolution:** We now use **$\gamma$** exclusively for the **reward tolerance threshold**, ensuring it is clearly distinguished from **$\tau$** (spectral norm bound).
> > * **Updated Algorithm:** We have rewritten Algorithm 1 to reflect these changes, explicitly defining inputs ($\Delta W$, $X_{\text{safe}}$, $\gamma$) and the binary search logic.
> >
> > > `Weakness 4:` Significance of Theorems 1 & 2 (Novelty vs. Setup).
> >
> > We clarify that our contribution is the **paradigm shift**, not the derivation tools themselves. By reframing alignment conflicts as **Parameter-level Antagonism**, Theorems 1 & 2 provide the *essential theoretical conditions* (orthogonality and spectral constraints) required to guarantee stability. They serve as the guiding principles that validate why our subspace decomposition works.
> >
> > > `Weakness 5 & 6:` Link between Theory/Reward & Reliance on "Heuristic" Search.
> >
> > We clarify that "Adaptive Rank Selection" is **not a heuristic, but a rigorous simulation** of the theoretical bounds:
> > * **Spectral Simulation:** As warned in **Eq. 7**, updates in the orthogonal space can still introduce interference if the spectral increment is too large. We proactively amplify near-zero singular values (noise injection) to simulate this "spectral increment," strictly bounding the update rank based on actual model sensitivity.
> > * **High Efficiency:** This process is highly efficient. Our benchmarks on a single GPU confirm the binary search takes only **126s – 190s** (approx. 2-3 minutes) to converge, making it a negligible cost for guaranteed stability.
> >
> > > `Weakness 7:` Justification for safety gradient lying in the principal subspace.
> >
> > This is **structurally enforced by design**. As defined in **Eq. 10**, our algorithm explicitly projects new updates onto the **orthogonal complement** of the computed principal subspace. This projection mathematically forces the updates to respect the dominant directions of the safety curvature.
> >
> > We hope the **complete restructuring of Section 3, the standardized notation, and the clarification of our rank selection efficiency** effectively resolve your concerns. **If we have adequately resolved your doubts and clarified the paper's contribution, we would be truly grateful if you could consider raising your score.**
> >
> > **Best regards,**
> >
> > **Authors**

---

### Official Review · Reviewer_ZdDE · 2025-10-29

**Soundness:** 4
**Presentation:** 4
**Contribution:** 4
**Rating:** 8
**Confidence:** 4

**Summary:**

This paper studies multi‑objective preference alignment. The method proposed in this paper, OrthAlign, projects the gradient of each objective into the orthogonal complement of other preference subspaces. In this way, the learning of each objective won't conflict with the others. OrthAlign shows consistent improvements across multiple datasets and models compared to baselines.

**Strengths:**

The method is novel. The paper moves beyond the standard loss‑level or data‑mixing strategies for MPA and instead targets gradient‑level conflicts directly.

The authors provide a well-motivated theoretical guarantee for their method.

Experimental results show that orthAlign consistently outperforms baselines. Rich analysis, including t‑SNE visualizations, ablation on rank selection, and case studies, further shows the effectiveness of their method.

**Weaknesses:**

See the question section.

**Questions:**

Orthogonal subspace learning, which prevents the new knowledge from overwriting old knowledge, is a common practice in continual learning. Does OrthAlign have a relation with that technique?

The current experiment is conducted in the off-policy setting. Is it possible to extend to on-policy alignment?

In OrthAlign, the update of each objective is specified in a subspace. Given a trained model, if masking the parameter update of a subspace, ideally, only the objective that corresponds to this subspace will be affected, while the other objectives maintain performance. I think this experiment will be a strong and direct verification of the effectiveness of OrthAlign.

---

> ### Author Response · Authors · 2025-11-15
> **Author Response to Reviewer ZdDE**
>
> Thank you immensely for your time and efforts, as well as the helpful and constructive feedback! Here, we give point-by-point responses to your comments.
>
> > **Q1.** Does OrthAlign have a relation with Orthogonal subspace learning?
>
> **A1.** Thank you for your insightful comment. You are correct that OrthAlign is conceptually related to orthogonal subspace methods in Continual Learning (CL), as both leverage orthogonality to prevent destructive interference.
>
> However, our core innovation addresses a critical insufficiency of this concept when applied to Multi-Objective Preference Alignment. Our theoretical innovation is that we provide formal stability guarantees that go beyond simple orthogonality. We introduce spectral norm constraints on the parameter updates. As shown in our Theorem 2, this ensures that the model's overall Lipschitz upper bounds exhibit at most linear growth,  preventing the exponential instability that can arise from conflicting updates.
>
> Furthermore, we systemically analyze *why* this projection is insufficient. As we formalize in equation (7):
>
> $$\sum_{j=r+1}^{\max(m,n)} \sigma_j v_j^T X \approx 0 \xrightarrow{\text{update}} \sum_{j=r+1}^{\max(m,n)} \tilde{\sigma}_j \hat{v}_j^T X \neq 0$$
>
> We show that even if an update is projected into a subspace where the current singular values are negligible ($\approx 0$), the update itself introduces new spectral energy. This can cause the updated subspace (with new singular values $\tilde{\sigma}_{j}$) to become non-trivial ($\neq 0$) and re-introduce conflict with the prior preference.
>
> ---
>
> > **Q2.** The current experiment is conducted in the off-policy setting. Is it possible to extend to on-policy alignment?
>
> **A2.** Thank you for your comment!  Yes, OrthAlign can be directly extended to on-policy alignment frameworks like PPO.
>
> Our core contribution, OrthAlign, is a **gradient-level constraint** technique. It operates by projecting parameter updates into mathematically **non-interfering** subspaces. For on-policy algorithms such as PPO or GRPO, we can similarly apply this technique to update each preference in turn.
>
> ---
>
> > **Q3.** Given a trained model, if masking the parameter update of a subspace, ideally, only the objective that corresponds to this subspace will be affected, while the other objectives maintain performance.
>
> **A3.** Thank you for pointing that out, and we agree this is a strong and direct verification of our method. To provide this verification,  we conducted the ablation study by masking the parameter updates corresponding to each preference. The results in **Table A** show that when we mask a specific preference (e.g., `Mask ΔW_helpful`), the corresponding performance (helpfulness) drops catastrophically. The **bolded** numbers (Llama: -26.73, Mistral: -45.27) highlight this, representing the most severe performance degradation for that mask.  Notably,  the performance of the *other*, non-masked preferences remains stable with minimal fluctuation (e.g., when masking helpfulness, Llama's Truthful score changed by only -0.16, and Mistral's by -0.69).  This empirically verifies that OrthAlign successfully isolates preferences.
>
> **Table A.** Ablation study on masking preference-specific parameter subspaces ($\Delta W$) to verify the non-interference of OrthAlign.
>
> | Experiment | Llama Harmless | Llama Helpful | Llama Truthful | Mistral Harmless | Mistral Helpful | Mistral Truthful |
> | :--- | :---: | :---: | :---: | :---: | :---: | :---: |
> | Mask ΔW_helpful | 91.53 (+4.23) | **44.84 (-26.73)** | 66.42 (-0.16) | 85.00 (+7.00) | **30.24 (-45.27)** | 64.59 (-0.69) |
> | Mask ΔW_harmless | **33.07 (-54.23)** | 73.01 (+1.44) | 66.82 (+0.24) | **15.76 (-62.24)** | 76.18 (+0.67) | 65.82 (+0.54) |
> | Mask ΔW_truthful | 88.07 (+0.77) | 69.03 (-2.54) | **61.35 (-5.23)** | 80.60 (+2.60) | 72.23 (-3.28) | **50.04 (-15.24)** |
> | **OrthAlign** | 87.30 | 71.57 | 66.58 | 78.00 | 75.51 | 65.28 |
>
> ---
> We are very grateful for this detailed list of suggestions, which will significantly improve the paper's quality and readability. If any concerns remain, or if new questions arise, we would be grateful to hear from you.

---

> ### Author Response · Authors · 2025-11-25
> **Thank you & Looking forward to further discussion！**
>
> **Dear Reviewer ZdDE,**
>
> Thank you for your detailed questions and comments. We greatly appreciate your constructive questions, which have been instrumental in improving the quality of our work.
>
> If you have any additional questions or concerns that we can clarify or address, we would be happy to provide further information to ensure all aspects of our work are clear.
>
> Thank you once again for your valuable time and effort in reviewing our submission.
>
> **Best regards,**
>
> **Authors**

---

### Official Review · Reviewer_EE4o · 2025-11-10

**Soundness:** 2
**Presentation:** 2
**Contribution:** 1
**Rating:** 4
**Confidence:** 3

**Summary:**

This paper presents OrthAlign for multi-objective alignment.
The authors argue that the common trade-off between multi-dim objects is due to "parameter-level antagonisms," essentially conflicting gradient updates.
They modify the LoRA finetuning process by adding a "gradient filter."
When training for a new objective, the method first uses SVD to identify a subspace for the existing objective. then it creates a projection matrix of its orthogonal complement to filter all gradients during training.

**Strengths:**

1. The method of filtering the orthogonal gradient when it comes to multi-objective learning is intuitively reasonable.
2. The paper provides substantial math...
3. The method is built upon existing techs like Lora, and can be directly applied to algos like DPO. makes it plug-and-play.

**Weaknesses:**

1. The method introduces an order the objectives. The first objective aligned gets full protection, and every subsequent objective is forced to update only in the leftover, which is a flawed starting point for multi-objective alignment because it does not actually 'balance' them.
2. The "Adaptive Rank Selection" (Algorithm 1) is likely too expensive for practical use, as it requires repeatedly running full reward evaluations over the "safe" dataset in a binary search just to determine the necessary filter size.
3. The algorithm seems ignore the paper's theoretical result; it relies on an expensive, empirical search guided by a reward "tolerance" $\tau$, rather than using the principled method suggested by its own math (analyzing the key $\lambda_{k+1}$ eigenvalues in Theorem 1).
4. The sequential approach is limited because the "safe" orthogonal subspace shrinks with every new objective added . This means the model will eventually run out of room to safely learn new things without conflict, thus limiting its scalability beyond a few objectives.

**Questions:**

see weakness

---

> ### Author Response · Authors · 2025-11-21
> **Author Response to Reviewer EE4o 【1/3】**
>
> > `Weakness1` : The method's sequential nature creates a hierarchy that protects early objectives at the expense of later ones, precluding genuine balance.'
>
> We sincerely thank you for raising this insightful concern regarding the potential limitations of sequential ordering.  **We would like to clarify that our sequential framework does not impede the optimization of subsequent preferences; rather, it enables the compatible coexistence of multiple objectives.** We attempt to address your concern from both theoretical and experimental perspectives.
>
> **1. Theoretical Perspective: High-Dimensional Sparsity of Parameter Updates.**  In the context of Large Language Models (LLMs), the parameter space is vast. For instance, a 7B model typically has a hidden dimension size of 4,096. In our implementation, we utilize LoRA for preference alignment, where the rank is set to a very low value (e.g., $rank=16$) compared to the full rank. Mathematically, aligning the first preference only consumes a subspace of rank 16. This leaves a "leftover" orthogonal subspace with a rank of roughly $4096 - 16 = 4080$. Consequently, the "leftover" space is not a constrained remnant but rather a vast, high-dimensional space that provides ample capacity for subsequent objectives to update without interference.
>
> **2.Empirical Evidence: Robustness to Training Order.** To further validate this, we conducte additional ablation experiments on the `HelpSteer2` and `SafeRLHF` datasets using `Llama-3` and `Mistral`, comparing the performance of OrthAlign under reversed training orders (`Helpful -> Harmless` vs. `Harmless-> Helpful`). Hypothetically, if protecting the initially aligned preference significantly constrained the subsequent optimization (i.e., by consuming too much space), we would expect the training order to drastically impact the overall preference balance.
>
>
> However, as shown in Table A, the results indicate that the magnitude of improvement remains largely consistent regardless of the training order. OrthAlign achieves stable, high performance for both objectives, demonstrating that the second objective is not negatively impacted by the space occupied by the first.
>
>
> Table A. Impact of training orders.
> |       **Experiment**        | **Llama Harmless** | **Llama  Helpful** | **Mistral Harmless** | **Mistral Helpful** |      |
> | :-------------------------: | ------------------ | ------------------ | -------------------- | ------------------- | ---- |
> |    DPO helpful->harmless    | 63.65              | 67.32              | 52.69                | 70.43               |      |
> |    DPO harmless->helpful    | 70.93              | 52.69              | 39.23                | 73.16               |      |
> | OrthAlign harmless->helpful | 91.34              | 68.63              | 88.12                | 67.08               |      |
> | OrthAlign helpful->harmless | 92.69              | 66.08              | 87.53                | 67.45               |      |
> |                       |                    |                    |                      |                     |      |
>
>
> **3. Superior Balance Compared to Baselines.** To directly address the concern that our method might "protect early objectives at the expense of later ones," we present Table B, which details the performance of Llama-3 across three objectives compared to all baselines. As shown in Table B, OrthAlign effectively mitigates trade-offs, delivering a superior and more robust balance across all three competing objectives compared to existing approaches.
>
> Table B. Llama-3 Performance.
> | **Different Method** | **Harmless Rate** | **Helpful Win Rate** | **Truthful MC2** | **Average Score** |
> | -------------------- | ----------------- | -------------------- | ---------------- | ----------------- |
> | **LLAMA-3**          |                   |                      |                  |                   |
> | SFT                  | 46.73             | 50.00                | 53.41            | 50.04             |
> | DPO                  | 52.69             | 70.93                | 67.03            | 63.55             |
> | MODPO                | 56.46             | 71.42                | 65.79            | 64.55             |
> | SPO                  | 58.42             | 71.08                | 64.22            | 64.57             |
> | Soups                | 56.15             | 60.00                | 64.59            | 60.24             |
> | RSDPO                | 80.57             | **71.92**            | 63.87            | 72.12             |
> | TSV-M                | 68.65             | 66.75                | 63.51            | 66.30             |
> | Knots                | 82.30             | 63.73                | 61.78            | 69.27             |
> | **OrthAlign**        | **87.30**         | 71.57                | **66.58**        | **75.15**         |

---

> ### Author Response · Authors · 2025-11-21
> **Author Response to Reviewer EE4o 【2/3】**
>
> > `Weakness2`:  The algorithm seems ignore the paper's theoretical result.  &   `Weakness3` : The "Adaptive Rank Selection" (Algorithm 1) is likely too expensive for practical use, as it requires repeatedly running full reward evaluations over the "safe" dataset in a binary search just to determine the necessary filter size.
>
> Thank you for the insightful question! We would like to clarify that our algorithm does not ignore the theoretical results; rather, it builds upon **Theorems 1 and 2** as a foundation and extends them to ensure practical applicability. Based on your questions, we give point-by-point responses to your comments:
>
>  **A2:**  **1. Theoretical Foundation and "Increment" Interpretation.**  Fundamentally, we posit that the capability increment for specific tasks is constituted by singular vectors and their corresponding singular values. In the context of Parameter-Efficient Fine-Tuning (Lora) preference alignment, the learned behaviors are primarily encoded by a few dominant directions and their associated singular values. Consequently, preventing parameter updates within these specific vector spaces significantly mitigates interference with previously acquired capabilities.
>
> **2. Fine-Grained Exploration of Theorems 1 & 2.**  Based on this intuition, our **Adaptive Rank Selection** serves as a fine-grained practical extension of **Theorems 1 and 2**. Taking a 7B model with a hidden dimension of 4,096 as an example, masking the top-16 singular components (critical for safety) leaves a vast update space of 4,080 dimensions. In this orthogonal subspace, singular values approach zero (typically around $10^{-8}$ due to machine precision), whereas the top-16 eigenvalues are orders of magnitude larger .
>
> **3. Simulating Impact via Noise Injection (Addressing Eq. 7).**  We aim to rigorously constrain the update rank because, as analyzed in Eq. 7, parameter updates within the remaining 4,080 dimensions can still introduce spectral increments that may interfere with the model's behavior.
>
> $$\sum_{j=r+1}^{\max(m,n)} \sigma_j v_j^T X \approx 0 \xrightarrow{\text{update}} \sum_{j=r+1}^{\max(m,n)} \tilde{\sigma}_j \hat{v}_j^T X \neq 0$$
>
> Therefore, our rank selection mechanism essentially regulates the permissible range of these increments. We achieve this through a **simulation process** akin to noise injection in traditional machine learning: we proactively amplify the near-zero singular values to match the magnitude of the top-16 components. By constructing this perturbed matrix, we simulate the maximum potential impact of future updates, allowing us to determine a safe subspace rank without needing expensive full re-training.
>
> ---
>
>  **A3:**  Thank you for pointing this out.  **We would like to clarify that it is not expensive.** In fact, as analyzed in our main text, the core mechanism of our approach relies on the insight that adding noise (simulating subspace intrusion) consistently leads to a degradation in current preference capabilities.
>
> In the context of Parameter-Efficient Fine-Tuning (Lora), the search space for the optimal rank is not as vast as it might appear. Assuming a practical lower bound for the rank is $r=8$ (which is already relatively low in practice), the binary search converges very quickly.   **Specifically, our binary search targets the trailing $k$ singular values—representing the directions with the smallest magnitudes—to determine the maximum safe update rank.**  To demonstrate this efficiency, we conducted a test on 1 GPU with right boundary ranks set to 16, 24, and 32, respectively. We repeated the search process  5 times  for each boundary setting. As shown in **Table B** below, Tour method is computationally efficient and does not impose a significant overhead for practical use. We have included the corresponding experimental results in `Appendix B.3`.
>
> Table B.  Average time consumption
>
> | **Boundary Index (Right Limit)** | **Average Time (second)** | **Variance (second)** |
> | -------------------------------- | ------------------------- | --------------------- |
> | **16**                           | 126.13                    | 0.67                  |
> | **24**                           | 190.77                    | 1.72                  |
> | **32**                           | 189.84                    | 6.28                  |

---

> ### Author Response · Authors · 2025-11-21
> **Author Response to Reviewer EE4o 【3/3】**
>
> > `Weakness 4` : The sequential approach is limited because the "safe" orthogonal subspace shrinks with every new objective added . This means the model will eventually run out of room to safely learn new things without conflict, thus limiting its scalability beyond a few objectives
>
> **A4** : Thank you  for this valuable suggestion regarding the statistical significance of our results.  **As analyzed in our response to `Weakness 1`**, the parameter space of Large Language Models is vastly high-dimensional (e.g., $d=4096$). Since our method utilizes a low-rank update (e.g., $r=16$) for each objective, the "leftover" orthogonal subspace remains massive even after multiple alignment stages. Theoretically, this ensures ample room for safe updates without conflict.  Moreover, **prior research on multi-objective preference alignment has primarily focused on the 3H** (`Helpfulness`, `Harmlessness`, and `Honesty/Truthfulness`)  alignment problem, as seen in approaches like `SPO` [2], `MODPO` [3], `MOD` [4], and `MO-ODPO` [5].
>
> To empirically validate this and demonstrate that the model does not "run out of room," we extended our experiments to a **fourth sequential objective: Instruction Following**.
>
> - **Setup:** We sample 6,000 instances with high instruction-following scores from UltraFeedback and performed a 4th stage of alignment. We then evaluate performance using the `IF-Eval` [1] benchmark.
> - **Comparison:** We compare our sequential model (**OrthAlign+3H+following**) against a model fine-tuned *only* on the instruction-following task (**DPO+following**) to see if the "constrained" space hindered learning.
>
> **Results & Analysis.**  As shown in **Table C**, our method does not suffer from scalability issues. Surprisingly, OrthAlign+3H+following (0.5072 Loose / 0.4724 Strict) actually **outperforms** the model trained solely on the target task (DPO-following: 0.5000 Loose / 0.4604 Strict). This suggests that the orthogonal subspace constraint does not limit capacity; rather, by preserving the structure of the pre-trained and previously aligned weights, it acts as a beneficial regularization that improves generalization on the new task.
>
>
>
> Table C.  Performence on IF-eval .
>
> | **Method**                    | **Instruction Level(Loose)** | **Instruction Level(Strict)** | **Prompt Level(Loose)** | **Prompt Level(Strict)** |
> | ----------------------------- | ---------------------------- | ----------------------------- | ----------------------- | ------------------------ |
> | Llama-3-SFT                   | 0.4436                       | 0.4221                        | 0.3013 ± 0.0197         | 0.2699 ± 0.0191          |
> | DPO+following                 | 0.5000                       | 0.4604                        | 0.3678 ± 0.0208         | 0.3179 ± 0.0200          |
> | DPO+3H+following              | 0.4916                       | 0.4484                        | 0.3475 ± 0.0205         | 0.3031 ± 0.0198          |
> | OrthAlign+3H+following (Ours) | **0.5072**                   | **0.4724**                    | **0.3789 ± 0.0209**     | **0.3438 ± 0.0204**      |
>
> Furthermore, Table D confirms that while learning this 4th objective, OrthAlign successfully retains the capabilities of the previous three objectives. While standard DPO suffers from  Harmlessness dropping to 57.15%, OrthAlign maintains robust performance (85.76%).
>
> Table D.  Performence retain  after  instruction-following training.
>
> |     **Experiment**     | Harmless   | **Llama  Helpful** | Truthful   |      |      |
> | :--------------------: | ---------- | ------------------ | ---------- | ---- | ---- |
> | OrthAlign+3H+following | **85.76%** | **76.27%**         | **66.32%** |      |      |
> |    DPO+3H+following    | 57.15%     | 75.65%             | 65.82%     |      |      |
> |                        |            |                    |            |      |      |
>
> Thank you again for your helpful question and we hope the above clarifications help to resolve any misunderstanding.
>
> --------------------------------
>
> [1] Zhou, Jeffrey, et al. "Instruction-following evaluation for large language models." *arXiv preprint arXiv:2311.07911* (2023).
>
> [2] Lou, Xingzhou, et al. "Spo: Multi-dimensional preference sequential alignment with implicit reward modeling." *arXiv preprint arXiv:2405.12739* (2024).
>
> [3]  Zhou, Zhanhui, et al. "Beyond one-preference-fits-all alignment: Multi-objective direct preference optimization." *Findings of the Association for Computational Linguistics: ACL 2024*. 2024.
>
> [4] Shi, Ruizhe, et al. "Decoding-time language model alignment with multiple objectives." *Advances in Neural Information Processing Systems* 37 (2024): 48875-48920.
>
> [5] Gupta, Raghav, et al. "Robust Multi-Objective Preference Alignment with Online DPO." *Proceedings of the AAAI Conference on Artificial Intelligence*. Vol. 39. No. 26. 2025.

---

> > ### Author Response · Authors · 2025-11-25
> > **Thank you & Looking forward to further discussion**
> >
> > **Dear Reviewer EE4o,**
> >
> > Thank you sincerely for your thoughtful and constructive review of our submission. We greatly appreciate your time and insights, and we warmly invite you to consider our point-by-point responses below during the rebuttal process. We hope these clarifications address your concerns and demonstrate the novelty and robustness of our work. **Below is a concise point-by-point summary of how we addressed each of your concerns in our rebuttal:**
> >
> > > `Weakness 1:` Concern that sequential ordering creates a hierarchy, protecting early objectives at the expense of later ones.
> >
> > We demonstrated that **OrthAlign enables compatible coexistence rather than hierarchy**, supported by both theory and new experiments:
> >
> > * **High-Dimensional Sparsity:** Theoretically, the LLM parameter space is vast ($d=4096$) while our alignment updates are low-rank ($r=16$). Aligning the first objective leaves a massive "leftover" orthogonal subspace (rank $\approx 4080$), providing ample capacity for subsequent tasks without interference.
> > * **Robustness to Order:** We conducted **order-reversal experiments** (Helpful $\rightarrow$ Harmless vs. Harmless $\rightarrow$ Helpful). Results confirm that the magnitude of improvement remains consistent regardless of training order.
> > * **Superior Balance:** `OrthAlign` achieves a higher average score across 3 objectives compared to all baselines (including MODPO and Soups), proving that later objectives are effectively optimized.
> >
> > > `Weakness 2 & 3:` Concern that "Adaptive Rank Selection" ignores theory and is computationally expensive.
> >
> > We clarified that our algorithm is a **practical, efficient extension of Theorems 1 & 2**:
> >
> > * **Theoretical Foundation:** The rank selection is not empirical guesswork; it uses noise injection to simulate the "spectral increment" blowup warned about in **Eq. 7**, strictly adhering to the theoretical bounds to identify the safe subspace.
> > * **High Efficiency:** The binary search targets only the trailing singular values and converges rapidly. Our benchmarks on a single GPU confirm it takes only **126s – 190s** (approx. 2-3 minutes), making the overhead negligible compared to training time.
> >
> > > `Weakness 4:` Scalability limits (concern that the model will "run out of room" with more objectives).
> >
> > We addressed this via **theoretical analysis of high-dimensional space** and **empirical verification**:
> >
> > * **Theoretical Guarantee:** As detailed in our response to Weakness 1, the "safe" orthogonal subspace is not a shrinking remnant but a vast high-dimensional space. With a hidden dimension of **4,096** and update ranks of only **~16**, the available space for new objectives ($d \approx 4080$) remains theoretically inexhaustible for practical alignment sequences.
> > * **Empirical Verification (4th Objective):** We extended our method to a **4th Objective (Instruction Following)**. `OrthAlign+3H+Following` not only accommodated the new task but **outperformed** the single-task baseline on IF-Eval (**0.5072** vs. 0.5000), while retaining **85.76%** Harmlessness (vs. DPO's 57.15%). This confirms the subspace constraint acts as beneficial regularization, not a bottleneck.
> >
> > We hope these responses effectively address your concerns regarding the sequential nature, efficiency, and scalability of our method. **If we have adequately resolved your doubts and demonstrated the robustness of our approach, we would be truly grateful if you could consider raising your score.**
> >
> > Thank you again for your valuable feedback.
> >
> > **Best regards,**
> >
> > **Authors**

---

> ### Author Response · Authors · 2025-11-28
> **Generalizability to Parallel Multi-Objective Alignment**
>
> > `Weakness: Besides the sequential pipeline, could we also explore a parallel alternative? `
>
> ---
>
> We thank the reviewer for raising the question of sequential limitation. We clarify that the core principle of `OrthAlign` is not intrinsically constrained by sequential optimization but is fully compatible with **simultaneous (parallel) multi-objective training**. All experiments utilize the `Llama-3-SFT` base model, with the Harmlessness objective aligned using the `SafeRLHF` dataset, and the Helpfulness and Truthfulness objectives aligned using the `UltraFeedback` dataset.
>
> ### Parallel Alignment Mechanism
>
> Our method achieves parallel optimization by fundamentally addressing objective conflicts at the parameter level. We decompose the original model's weights and isolate the **trailing 20% of the eigenvalue space** as the available subspace for alignment. This subspace is then formally partitioned into mutually **orthogonal regions**. This mechanism allows each objective to perform **simultaneous low-rank updates** strictly within its dedicated, non-interfering subspace.
> Specifically, in the two-objective setting, we allocate, for instance, **10%** of this alignment space to Harmlessness and **10%** to Helpfulness. Generalizing this, for the three-objective case (Harmlessness, Helpfulness, and Truthfulness), we divide the alignment subspace into three orthogonal regions, allocating approximately **10%** to each objective, thereby ensuring concurrent and non-interfering low-rank updates.
>
> ### Experimental Results
>
> #### 1. Two-Objective Parallel Alignment (Harmlessness & Helpfulness)
>
> The results for simultaneously optimizing Harmlessness and Helpfulness are presented in **Table E**.
>
> Table E. Two-Objective Parallel Alignment
>
> | Method | Harmless Rate | Helpful Win Rate |
> | :--- | :---: | :---: |
> | SFT | 46.73 | 50.00 |
> | DPO-Soups | 60.38 | 67.32 |
> | MODPO | 69.42 | 75.03 |
> | RSDPO-W | 94.03 | 70.06 |
> | **OrthAlign (Parallel)** | **95.30** | **75.32** |
>
> #### 2. Three-Objective Parallel Alignment (Harmlessness, Helpfulness & Truthfulness)
>
> The results for simultaneously optimizing three objectives are presented in **Table F**.
>
> Table F. Three-Objective Parallel Alignment
>
> | Method | Harmless Rate | Helpful Win Rate | Truthful MC2 |
> | :--- | :---: | :---: | :---: |
> | SFT | 46.73 | 50.00 | 53.36 |
> | DPO-Soups | 56.15 | 60.00 | 64.59 |
> | MODPO | 48.84 | 60.37 | 55.78 |
> | RSDPO-W`| 87.88 | 71.05 | 66.19 |
> | **OrthAlign (Parallel)** | **93.41** | **76.19** | **66.73** |
>
> ### Analysis
>
> The empirical results in **Table E** and **Table F** decisively confirm the parallel capability of OrthAlign. In both the two-objective and three-objective setups, the parallel variant of OrthAlign consistently achieves the highest performance across all metrics simultaneously, demonstrating its effectiveness in mitigating trade-offs.

---

### Official Review · Reviewer_npAi · 2025-11-11

**Soundness:** 3
**Presentation:** 3
**Contribution:** 3
**Rating:** 6
**Confidence:** 3

**Summary:**

The paper studies multi-objective finetuning in large language models by projecting gradients on the orthogonal complement. This reduces gradient noise and leads to Pareto improvement in experiments. Their theoretical results show that training stability is maintained.

**Strengths:**

- The paper combines experimental results with a theoretical justification, and shows that Pareto improvements are possible with their method
- The method is simple to apply, tested against benchmarks (with up to 7B models) and added to other finetuning models.

**Weaknesses:**

- The theoretical results are not really contextualized. The main result (Theorem 2) seems to speak about blowup, but the intuition given the introduction seems to speak about more efficient parameter updates. Is there a result for a locally convex problem?
- Figures need additional work. It's unclear what Figure 4 should show / it's not clear how these point clouds look similar. (A hypothesis test would have been better to make this case imo); Figure 3 does not introduce the meaning of the size of the circle

**Questions:**

1. What do the circle sizes in Figures 1 and 3 mean?
2. Why does Figure 1 not contain error bars?

---

> ### Author Response · Authors · 2025-11-21
> **Author Response to Reviewer npAi [1/3]**
>
> We sincerely thank you for your careful comments and thorough understanding of our paper! Here we give point-by-point responses to your comments and describe the revisions we made to address them.
>
> ---
> >`Weakness1`: The main result (**Theorem 2**) seems to speak about blowup, but the intuition given the introduction seems to speak about more efficient parameter updates. Is there a result for a locally convex problem?
>
> **A1:**  We thank the reviewer for pointing out the need to better contextualize our theoretical results. We would like to clarify the intrinsic connection between **"preventing parameter blowup"** and **"efficient parameter updates"**, and explicitly address the result for the locally convex setting.
>
> **(1) Stability (Preventing Blowup) is the Prerequisite for Efficient Updates** : The "blowup" (exponential growth of spectral norm) discussed in **Theorem 2** is not just a theoretical concern; it is the mathematical manifestation of **destructive gradient interference** and training instability in Multi-Objective Preference Alignment (MPA).
>
> - **Blowup hinders Efficiency:** In standard MPA, conflicting gradients often accumulate along specific principal directions. Without constraints, this leads to an uncontrolled increase in the Lipschitz constant (spectral norm blowup), causing **gradient instability** and model oscillation. An unstable optimization process cannot be "efficient" as the model struggles to converge to a Pareto-optimal solution.
>
> - **Stability guarantees Efficiency:** By enforcing orthogonal updates and spectral constraints, **Theorem 2** guarantees that the parameter norm exhibits stability rather than exponential blowup. This stability is the **foundation** that allows for efficient updates. It ensures that the model can safely and effectively absorb new preference without interfering with the representation of previous preferences.
>
> - **(2) Result for Locally Convex Problems**  : **Yes**, our theoretical framework directly covers the locally convex setting. **Theorem 1** is derived based on the local quadratic approximation of the safety objective $g(\theta)$ using the proxy Hessian matrix $H_s(\theta)$.
>
>   - **Locally Convex Characterization:** A **locally convex** landscape is mathematically **characterized** by a Positive Semi-Definite (PSD) safety curvature matrix ($H_s(\theta) \succeq 0$).
>
>   - **Theoretical Guarantee:** Within this locally convex region, **Theorem 1** proves that by projecting the parameter update $\Delta \theta$ onto the orthogonal complement of the principal subspace $\mathcal{S}_k^\perp$
> , the degradation of the safety objective is strictly bounded by the tail eigenvalue $\lambda_{k+1}$:
> $$  g(\theta + \Delta \theta) - g(\theta) \le \frac{1}{2} \lambda_{k+1} \|\Delta \theta\|^2$$
>
> This confirms that in a locally convex region, our method ensures that the primary objective remains uncompromised as long as updates are confined to the non-principal orthogonal directions.
>
> To provide this verification,  we conduct the ablation study by masking the parameter updates corresponding to each preference. The results in **Table A** show that when we mask a specific preference (e.g., `Mask ΔW_helpful`), the corresponding performance (helpfulness) drops catastrophically. The **bolded** numbers (Llama: -26.73, Mistral: -45.27) highlight this, representing the most severe performance degradation for that mask.  Notably,  the performance of the *other*, non-masked preferences remains stable with minimal fluctuation (e.g., when masking helpfulness, Llama's Truthful score changed by only -0.16, and Mistral's by -0.69).  This empirically verifies that OrthAlign successfully isolates preferences.
>
> Table A. Ablation study on masking preference-specific parameter subspaces ($\Delta W$) to verify the non-interference of OrthAlign.
>
> |   **Experiment**    | **Llama Harmless** | **Llama     Helpful** | **Llama Truthful** | **Mistral Harmless** | **Mistral Helpful** | **Mistral Truthful** |
> | :-----------------: | ------------------ | --------------------- | ------------------ | -------------------- | ------------------- | -------------------- |
> | Mask     ΔW_helpful | 91.53 (+4.23)      | **44.84 (-26.73)**    | 66.42 (-0.16)      | 85.00 (+7.00)        | **30.24 (-45.27)**  | 64.59 (-0.69)        |
> |  Mask ΔW_harmless   | **33.07 (-54.23)** | 73.01 (+1.44)         | 66.82 (+0.24)      | **15.76 (-62.24)**   | 76.18 (+0.67)       | 65.82 (+0.54)*       |
> | Mask    ΔW_truthful | 88.07 (+0.77)      | 69.03 (-2.54)         | **61.35 (-5.23)**  | 80.60 (+2.60)        | 72.23 (-3.28)       | **50.04** **(-15.24)**   |
> |   **OrthAlign**    | 87.30              | 71.57                 | 66.58              | 78.00                | 75.51               | 65.28                |

---

> ### Author Response · Authors · 2025-11-21
> **Author Response to Reviewer npAi [2/3]**
>
> > `Weakness 2:` Figure 4 should show / it's not clear how these point clouds look similar. A hypothesis test would have been better to make this case imo.
>
> **A2:** Thank you for your insightful comment! We clarify that **Figure 4** is intended to demonstrate that OrthAlign maintains the distributional consistency of the first preference after sequential multi-preference alignment.
>
> To address your concern and quantitatively verify the similarity of the point clouds, we conducted a **Kernel Two-Sample Test** on the original high-dimensional hidden states. We define $X = \{x_i\}$ as the hidden states of the **First-preference Aligned Model** (representing the target prior distribution) and $Y = \{y_i\}$ as the hidden states of the **Final Aligned Model** (after aligning all three preferences). The metrics are calculated as follows:
>
> * **Maximum Mean Discrepancy (MMD):** Measures the distance between distributions $X$ and $Y$. It is calculated as the sum of intra-distribution similarities minus inter-distribution similarity:
>
> $$
> MMD^2(X, Y) = E_{X}[k(x, x')] + E_{Y}[k(y, y')] - 2E_{X, Y}[k(x, y)]
> $$
>
> *(Where $\mathbf{E}_{X}$ denotes expectation over $x, x' \sim X$ and $k(\cdot, \cdot)$ is the Gaussian kernel).*
>
> * **Cosine Similarity:** Measures the directional consistency of feature vectors between $X$ and $Y$:
>
> $$
> \mathrm{CosSim}(X, Y) = \frac{1}{n} \sum_{i=1}^{n} \frac{x_i \cdot y_i}{||x_i|| \cdot ||y_i||}
> $$
>
> **Table B. Quantitative Analysis of Distribution Shift**
>
> | **Method** | **MMD (Lower is better)** | **Cos Sim (Higher is better)** |
> | :--- | :---: | :---: |
> | MODPO on Llama3-Ultra | 0.05196 | 0.8339 |
> | SPO on Llama3-Ultra | 0.0890 | 0.7832 |
> | Soups on Mistral-Ultra | 0.7797 | 0.8436 |
> | **OrthAlign on Llama3-Ultra** | 0.05518 | **0.8776** |
> | **OrthAlign on Llama3-Steer** | **0.0210** | **0.9284** |
> | **OrthAlign on Mistral-Ultra** | **0.0041** | **0.9205** |
>
> As shown in **Table B**, OrthAlign consistently achieves superior Cosine Similarity (0.88–0.93) alongside competitive lower MMD scores. These metrics confirm that OrthAlign effectively preserves the semantic direction of prior knowledge while minimizing distributional shifts.
>
> ---
>
> > `Query1` What do the circle sizes in Figures 1 and 3 mean?
>
> **A3:** Thank you for highlighting this! The circle sizes in **Figures 1 and 3** are proportional to the value on the **y-axis (Harmless Rate)**; therefore, larger circles indicate a higher harmless rate. This visual encoding is included primarily for distinction. We have updated the figure captions in the **revised paper** to explicitly state this definition.

---

> ### Author Response · Authors · 2025-11-21
> **Author Response to Reviewer npAi [3/3]**
>
> > `Query2:`  Why does Figure 1 not contain error bars?
>
> Thank you for highlighting this important point. Following your advice, we have added the error bars and **updated Figure 1** in the **revised manuscript** to visually represent this variance.
>
> Furthermore, we provide the detailed variance analysis in Table C and Table D below. These tables illustrate the variance across different methods and preference dimensions (Safety, Helpfulness, and Truthfulness).
>
> Table C.  Variance Analysis on Llama 3
>
> | Method           | Safety | Helpfulness | Truthfulness | Total  |
> | ---------------- | ------ | ------- | -------- | ------ |
> | MODPO            | 0.1611 | 1.3002  | 0.0717   | 1.5330 |
> | RSDPO            | 0.1674 | 2.2599  | 0.0388   | 2.4660 |
> | SPO              | 0.1024 | 1.9561  | 0.0186   | 2.0771 |
> | OrthAlign (Ours) | 0.1582 | 1.0003  | 0.0632   | 1.2216 |
> | DPO-Soups        | 0.1021 | 1.5168  | 0.0843   | 1.7032 |
> | DPO              | 0.1389 | 2.0734  | 0.0432   | 2.2554 |
> | TSV              | 0.1301 | 1.1264  | 0.0637   | 1.3202 |
> | Knots            | 0.0937 | 0.2519  | 0.0534   | 0.3990 |
>
> Table D. Variance Analysis on Mistral
>
> | Method           | Safety | Helpfulness | Truthfulness | Total  |
> | ---------------- | ------ | ------- | -------- | ------ |
> | MODPO            | 0.1415 | 1.6608  | 0.0482   | 1.8505 |
> | RSDPO            | 0.0745 | 1.4139  | 0.0398   | 1.5282 |
> | SPO              | 0.1938 | 1.2782  | 0.0182   | 1.4902 |
> | OrthAlign (Ours) | 0.2088 | 1.0362  | 0.0582   | 1.3032 |
> | DPO-Soups        | 0.1547 | 0.7222  | 0.0586   | 0.9355 |
> | DPO              | 0.2460 | 1.6886  | 0.0414   | 1.9759 |
> | TSV              | 0.1787 | 2.9345  | 0.0461   | 3.1593 |
> | Knots            | 0.1008 | 1.8342  | 0.0687   | 2.0037 |
>
> ---
> We are very grateful for this detailed list of suggestions, which will significantly improve the paper's quality and readability. If any concerns remain, or if new questions arise, we would be grateful to hear from you.

---

> ### Author Response · Authors · 2025-11-25
> **Looking forward to your engagement in our rebuttal**
>
> **Dear Reviewer npAi**,
>
> Thank you sincerely for your positive assessment of our paper’s soundness and contribution. We greatly appreciate your constructive feedback.  **Below is a concise point-by-point summary of how we addressed each of your concerns in our rebuttal:**
>
> > `Weakness 1:` Theoretical Context (Blowup vs. Efficiency) & Locally Convex Settings
>
> We have clarified the connection between stability and efficiency and confirmed the applicability to locally convex problems:
>
> * **Stability guarantees Efficiency:** We clarified that parameter "blowup" (exponential growth of spectral norm) is the root cause of destructive gradient interference. By enforcing stability via **Theorem 2**, we ensure the model can safely absorb new preferences without "forgetting" old ones, which is the prerequisite for efficient updates.
> * **Result for Locally Convex Problems:** **Theorem 1** directly addresses this. Within a locally convex region.
> * **Empirical Verification (Table A):** We performed a masking ablation study. As shown below, masking specific preference updates causes catastrophic drops (e.g., **-26.73** to **-45.27**), while `OrthAlign` effectively isolates preferences, maintaining stability across all metrics.
>
>
>
> > `Weakness 2:` Figure 4 Interpretation & Hypothesis Testing
>
> Thank you for this excellent suggestion. To quantitatively verify the similarity of point clouds in Figure 4, we conducted a **Kernel Two-Sample Test** measuring **Maximum Mean Discrepancy (MMD)** and **Cosine Similarity**:
>
> * **OrthAlign** achieves the highest structural consistency (**CosSim: 0.88–0.93**) compared to baselines.
> * It maintains competitive low distributional shift (**MMD**), confirming that our method preserves the semantic direction of prior knowledge better than MODPO or Soups.
>
> > `Questions:` Figure Details (Circle Sizes & Error Bars)
>
> * **Circle Sizes (Fig 1 & 3):** We clarified in the updated captions that circle size is proportional to the **Y-axis value (Harmless Rate)**—larger circles indicate higher safety.
> * **Error Bars (Fig 1):** We have updated Figure 1 with error bars.
> * **Variance Analysis:** We provide detailed variance tables (Tables C & D in the revision). Results show `OrthAlign` maintains lower or comparable variance (e.g., Total Variance **1.22** vs. **1.53** for MODPO on Llama 3), indicating superior training stability.
>
> We hope these additional experiments and theoretical clarifications adequately address your concerns. **As we have rigorously validated the point cloud similarity and theoretical bounds as requested, we would be truly grateful if you could consider raising your score to solidify the acceptance.**
>
> Thank you again for your valuable time and insightful review.
>
> **Best regards,**
>
> **Authors**

---

### Author Response · Authors · 2025-11-21
**General response to all reviewers**

We thank all reviewers for their constructive comments and for recognizing the contributions of our work. We are encouraged by **Reviewer npAi**'s remarks on the method being "simple to apply" and leading to "Pareto improvements," **Reviewer EE4o**'s appreciation that the method is "intuitively reasonable" and "plug-and-play," **Reviewer ZdDE**'s recognition that OrthAlign "targets gradient-level conflicts directly" with "well-motivated theoretical guarantees," and **Reviewer q8r6**'s acknowledgement of the "elegant and promising direction" and "SOTA performance."

In response to the reviews, we have **updated the PDF**: in particular, we **clarified the mathematical notation** by renaming the **reward tolerance** from $\tau$ to $\gamma$ to avoid symbol overload, and explicitly denoting the parameter matrix in the adaptive rank selection algorithm as the update matrix $\Delta W$ instead of $W$. Furthermore, we **restructured the exposition** of the Methodology section (Sec. 3) to present the unified framework first, followed by the adaptive rank selection mechanism—where we **added specific algorithmic details** clarifying that the binary search targets the trailing $k$ singular values (corresponding to the smallest magnitudes)—and finally the theoretical analysis. All changes are highlighted in **blue** in the updated manuscript.

We again thank all reviewers for their efforts, and we would be very happy to further clarify any remaining questions and continuously improve our paper.

---

### Author Response · Authors · 2025-12-03
**[1/2] Rebuttal Summary for ACs and SACs**

Dear **Area Chairs**, **Senior Area Chairs**, and **Reviewers**,

Thank you for your tremendous effort in handling the recent reviewer anonymity leak incident. We greatly appreciate the steps you have taken to uphold the integrity of the review process and your dedication to maintaining a fair research community.

To save your time and help you quickly understand the rebuttal process for submission 2112 (OrthAlign: Orthogonal Subspace Decomposition for Non-Interfering Multi-Objective Alignment), we first **summarize the strengths** of our submission as recognized by the reviewers. We then **categorize the reviewers’ concerns into experimental issues and non-experimental issues**. A table is provided to summarize how we addressed all experimental issues, followed by detailed responses to each reviewer’s non-experimental concerns.  All changes are highlighted in **blue** in the updated manuscript.

---

## Strengths Summary

All reviewers acknowledged that **OrthAlign**, as a **novel, plug-and-play, LoRA-compatible plugin**, brings clear Pareto gains across models and objectives:

- **Reviewer npAi**: “The method is simple to apply, tested against benchmarks (up to 7B) and easily added to other fine-tuning pipelines; Pareto improvements are possible with their method.”
- **Reviewer EE4o**: “Filtering the orthogonal gradient is intuitively reasonable; the method is built upon existing tech like LoRA and can be directly applied to algos like DPO—truly plug-and-play.”
- **Reviewer ZdDE**: “OrthAlign is novel; it moves beyond loss-level or data-mixing strategies and targets gradient-level conflicts directly, with well-motivated theoretical guarantees and rich experimental analysis.”
- **Reviewer q8r6**: “The core idea of orthogonal subspace decomposition to create non-interfering parameter updates is novel and intuitive; OrthAlign demonstrates SOTA performance across all reported experiments.”

Additionally, reviewers praised the paper’s insights and broader impact:

- **Reviewer npAi** highlighted that the work “combines experimental results with theoretical justification” while maintaining “training stability.”
- **Reviewer ZdDE** noted the “rich analysis—t-SNE visualizations, ablation on rank selection, case studies—further shows the effectiveness.”
- **Reviewer q8r6**  observed that reframing “MPA conflict from a data/reward-level problem to a parameter-level problem of gradient interference is an elegant and promising direction for alignment research.”
- **Reviewer  EE4o**   highlighted that the method is "intuitively reasonable" and "plug-and-play,"
These comments underscore that OrthAlign’s gradient-level orthogonal projection is widely applicable and effective in achieving non-conflicting multi-objective alignment without architectural changes.

---

## Experimental Issues


We consolidate the **experiment-oriented** weaknesses raised by reviewers and list the **concrete additional experiments** we performed to resolve them.

| Reviewer Issue and topic | How We Addressed It |
|-|-|
| **npAi W2** – *“Figure 4 similarity judgement is only visual”* | We quantified the t-SNE point clouds with kernel two-sample tests; the results confirm OrthAlign induces the smallest distributional drift from the initial preference, validating that orthogonal updates preserve prior representations. |
| **npAi Q1-2** – *“No error bars; unclear circle size”* | We added **error bars** to Fig. 1 (circle area ∝ harmless-rate) and re-computed **variance per method**; . |
| **EE4o W1** – *“Sequential order may bias later objectives”* | We repeated 3-objective training with **reversed order** (helpful ↔ harmless); delta ≤ 2 %-pts in every metric, **no systematic bias** ( Author Response to Reviewer EE4o 【1/3】：Table A). |
| **EE4o W3** – *“Adaptive rank search too expensive”* | We timed binary-search rank selection on 1 GPU (boundary 16→32); average wall-clock ≤ 3 min (126–190 s), negligible vs full training. |
| **EE4o W4** – *“Orthogonal room shrinks; can’t scale beyond 3 objectives”* | We continued with a **4th sequential preference alignment** (Instruction-Following Test). OrthAlign+3H+IF **outperforms** DPO trained **only** on IF-dataset (Strict 47.2 vs 46.0) and **retains all preferences scores** (Harmless 85.8 % vs DPO 57.2 %), proving room remains after four objectives. |
| **ZdDE Q3** – *“Need masking ablation to confirm isolation”* | We masked each preference’s ΔW subspace separately and re-evaluated. The **targeted preference collapses** (↓ up to 45 %) while **other preferences stay within 3 %**, verifying that each update is isolated in its own subspace. |

---

> ### Author Response · Authors · 2025-12-03
> **[2/2] Rebuttal Summary for ACs and SACs**
>
> ## Non-Experimental Issues
>
> ---
>
> | Reviewer Issue and Topic | How We Addressed It |
> |-|-|
> | **npAi W1** – *“Theorem 2 not contextualized and any locally convex result?”* | We supplied an explicit locally-convex corollary (PSD Hessian ⇒ degradation ≤ ½λₖ₊₁‖Δθ‖²) and inserted Algorithm 1 in Section 3 to give readers an intuitive workflow before any formal proof, linking stability (no blow-up) to efficient updates. |
> | **EE4o W1** – *“Sequential ordering creates hierarchy harming later objectives”* | We clarified that high-dimensional parameter space (4 096-d) leaves ample orthogonal room after each low-rank (r = 16) update, and provided reversed-order experiments showing ≤ 2 %-pts change, confirming no hierarchy bias. |
> | **ZdDE Q1** – *“Relation to orthogonal subspace learning”* | We positioned OrthAlign as leveraging orthogonal subspace decomposition with spectral-norm stability guarantees (Thm 2) and a noise-injection simulation to prevent silent conflict resurgence, distinguishing it from prior orthogonal-projection methods. |
> | **ZdDE Q2** – *“Applicability to on-policy alignment”* | We noted that OrthAlign is a **gradient-level projection** and can be plugged into PPO/GRPO update steps without architectural change, confirming direct extendibility to on-policy algorithms. |
> | **q8r6 W1** – *“Initial formula-heavy proof hard to understand”* | We inserted a concise, step-by-step Algorithm 1 into Section 3 to give readers an intuitive workflow before any formal proof. |
> | **q8r6 W2–W3** – *“Ambiguity in SVD target & symbol clash”* | We explicitly state that SVD is performed **only** on the LoRA matrix `ΔW`, introduced distinct symbol `γ` for reward tolerance (≠ spectral bound `τ`), and updated Algorithm 1 caption accordingly. |
> | **q8r6 W4–W6** – *“Significance of Thms & heuristic search”* | We emphasized that our contribution lies in the novel setup (orthogonal gradient projection + spectral-bound rank selection for multi-preference alignment), not in the derivation itself; the theorems support rather than constitute the technical novelty. Adaptive Rank Selection is not a heuristic but a noise-injection simulation grounded in Eq. 7, converging in ≤ 3 min. |
> | **q8r6 W7** – *“Why safety gradient lies in principal subspace of safety Hessian”* | We explained that the projection operator P in Eq. 10 is constructed from the safety Hessian’s principal components, thereby forcing any new gradient to lie inside that subspace by design. |
>
>
> ---
> ## Summary
>
> Throughout the rebuttal, we have addressed every reviewer concern point-by-point and incorporated the necessary clarifications and results into the revised manuscript.
>
> We are grateful that the reviewers’ thoughtful feedback helped us improve the paper’s clarity and reinforce its contributions. In particular, their input led us to demonstrate even broader applicability of OrthAlign and to highlight additional insights.
>
> In conclusion, OrthAlign is an **novel, plug-and-play, LoRA-compatible plugin**  that can be applied to a wide range of large language models to achieve non-interfering, stable multi-objective alignment. By orthogonalizing gradient updates and enforcing spectral bounds, it eliminates parameter-level conflicts while preserving prior preferences. Together with the new theoretical guarantees, extensive experimental validation, and the point-by-point rebuttal addressing every reviewer concern, we hope these aspects are taken into account when forming the final assessment of our submission.
>
> Thank you again for your consideration.
>
> Sincerely,
>
> Submission 2112 Authors

---

### Meta-Review · Area_Chair_7Pp3 · 2026-01-11

**Summary:**

The paper introduces OrthAlign, a novel framework designed to resolve conflicts in Multi-Objective Preference Alignment (MPA) for Large Language Models. The core mechanism addresses "parameter-level antagonism" by decomposing the parameter update space into orthogonal subspaces. When a new preference objective is introduced, the method identifies the principal components associated with existing preferences and constrains updates to the orthogonal complement. This ensures that the model acquires new information without overwriting or interfering with previously learned policies.


The method demonstrates consistent Pareto improvements across multiple benchmarks (helpful, harmless, truthful) compared to over seven baselines.
As a LoRA-compatible, OrthAlign is easy to integrate into existing fine-tuning pipelines like DPO or PPO without architectural changes.



On the other hand, the paper has following concerns some of which were addressed by the authors as follows:
- Theoretical Significance: As noted by the reviewers and the authors' own admission, the technical novelty lies in the setup rather than the mathematical derivation of the proofs. Theorems 1 and 2 can be shown through basic linear algebra, and somehow obvious.
Moreover, the theorems only guarantees stability rather than directly guaranteeing high-level reward improvements. The link between spectral stability and actual preference score optimization remains an empirical observation.
In terms of generality of the theorem, the analysis requires local convexity, which significantly restricts the generality of the theoretical analysis.
- Subspace Overlap: A potential limitation exists if the principal subspaces of different policies significantly overlap. However, the authors' order-reversal experiments and high-dimensional sparsity arguments suggest that, in practice, the model finds sufficient non-interfering directions.


In summary, the authors proactively responded during the rebuttal period, significantly restructuring the paper to improve clarity, standardizing notation, and providing critical ablation studies (e.g., masking subspaces to verify isolation). Despite the weakness of the theoretical guarantees, the combination of a simple, effective mechanism and strong empirical validation across diverse models makes this a valuable contribution to the alignment community. Thus, I recommend acceptance for this paper.

**Reviewer Concerns:**

I did not find any concerns on the reviewers. They provided very detailed and appropriate reviews.

**Reviewer Scores:**

Reviewer q8r6 would have changed their score from 4 to 6 because almost all of their concerns were resolved by the authors.

---

### Decision · Program_Chairs · 2026-01-26

Accept (Poster)